# Test of Time: A Benchmark for Evaluating LLMs on Temporal Reasoning

**Bahare Fatemi**[1]*, **Mehran Kazemi**[2]*, **Anton Tsitsulin**[1], **Karishma Malkan**[2], **Jinyeong Yim**[3],
**John Palowitch**[2], **Sungyong Seo**[3], **Jonathan Halcrow**[1], **and Bryan Perozzi**[1]
[1]Google Research, [2]Google DeepMind, [3]Google

## Abstract

Large language models (LLMs) have showcased remarkable reasoning capabilities, yet they remain susceptible to errors, particularly in temporal reasoning tasks involving complex temporal logic. Existing research has explored LLM performance on temporal reasoning using diverse datasets and benchmarks. However, these studies often rely on real-world data that LLMs may have encountered during pre-training or employ anonymization techniques that can inadvertently introduce factual inconsistencies. In this work, we address these limitations by introducing novel synthetic datasets specifically designed to assess LLM temporal reasoning abilities in various scenarios. The diversity of question types across these datasets enables systematic investigation into the impact of the problem structure, size, question type, fact order, and other factors on LLM performance. Our findings provide valuable insights into the strengths and weaknesses of current LLMs in temporal reasoning tasks. The dataset is available at: https://huggingface.co/datasets/baharef/ToT.

## 1 Introduction

Recent breakthroughs in large language model (LLM) research and applications have been significant (Vaswani et al., 2017; Devlin et al., 2019; Raffel et al., 2020; Brown et al., 2020; Touvron et al., 2023; Achiam et al., 2023; Team et al., 2023; Reid et al., 2024). These models, capable of generating new content, have fascinated the AI community, leading to the release of numerous LLMs trained on diverse tasks and data types (Zhao et al., 2023). All of these advancements have led to a growing consensus that LLMs are a pivotal advancement on the path to artificial general intelligence (AGI) (Bubeck et al., 2023). Benchmarking reasoning capabilities in LLMs is therefore a problem of pressing interest to the field (Huang & Chang, 2023).

In this work, we focus on temporal reasoning, an essential task for intelligent systems across many domains. Temporal reasoning is focused on understanding reasoning between events in time. Despite this area's importance, existing temporal reasoning benchmarks do not effectively measure the full scope of temporal reasoning relationships. Instead, they typically rely on question-answering tasks based on Knowledge Graph (KG)-style temporal facts about well-known entities.

This overemphasis on KG-style temporal facts limits the scope of research and creates several issues. First, it neglects the diverse temporal structure and reasoning tasks found in real-world applications beyond KGs. Second, the results on such data often reflect a model's ability to exploit prior knowledge rather than genuine temporal reasoning, making findings less relevant to domains where models lack this knowledge (see Figure 1 as an example.). In addition, previous research has shown that shortcuts or heuristics can often answer questions on these datasets without explicit temporal reasoning (Chen et al., 2022; Tan et al., 2023). Finally, the simple temporal structure of knowledge graphs overlooks the extensive time arithmetic skills required in real-world temporal questions.

**Our Contributions**: To address these limitations, we develop tasks specifically designed to assess temporal reasoning in a more comprehensive and controlled manner. Our benchmark, Test of Time, `ToT`, is centered around the observation that temporal reasoning often involves two primary skills: 1) understanding the semantics and logic of time, and 2) the ability to carry out temporal arithmetic.

---

*Equal contribution. Correspondence: {baharef, mehrankazemi}@google.com.

**Prompt**: Below are the list of head coaches for Chelsea FC. Who was the coach before Pochettino?
Pochettino: July 2023 to May 2024
Potter: September 2022 to April 2023
Tuchel: January 2021 to September 2022
Lampard: July 2019 to January 2021 and April 2023 to June 2023
Sarri: July 2018 to June 2019

**Model Response**: The coach before Pochettino was **Frank Lampard** during his second stint with the club from April 2023 to June 2023.

**Prompt**: Below are the list of head coaches for a club. Who was the coach before E5?
E5: July 2023 to May 2024
E4: September 2022 to April 2023
E3: January 2021 to September 2022
E2: July 2019 to January 2021 and April 2023 to June 2023
E1: July 2018 to June 2019

**Model Response**: **E4** was the coach before E5.

Grounded prompt (answered correctly)   Abstract prompt (answered incorrectly)

Figure 1: Comparison of the same temporal query using real (left) and anonymized (right) entity names. Gemini Advanced correctly answered the query with real names but failed with anonymized names, suggesting that LLMs might rely on their parametric knowledge to solve temporal tasks.

`ToT` has two tasks to cover each essential skill of temporal reasoning, which enable measuring and improving model performances along these two axes independently. `ToT-Semantic`, a synthetic task, focuses on temporal semantics and logic; it allows for a flexible exploration of diverse graph structures and reasoning task complexity, isolating and evaluating reasoning abilities independent of prior knowledge. `ToT-Arithmetic`, a crowd-sourced task, assesses the ability to perform calculations involving time points and durations. Our experimental results with `ToT` provide valuable insights into the strengths and weaknesses of current LLMs in temporal reasoning tasks.

## 2 RELATED WORK

**Reasoning.** The ability to draw valid conclusions from explicitly provided knowledge has been a fundamental goal for AI since its early days (McCarthy, 1959; Hewitt, 1969). In the past few years, several LLM-based techniques have been developed which have advanced the general automated reasoning capabilities of the state-of-the-art models (Wei et al., 2022; Yao et al., 2023), or their capabilities in specific directions including mathematical reasoning (Lewkowycz et al., 2022; Ahn et al., 2024), logical reasoning (Creswell et al., 2022; Kazemi et al., 2023b), multi-modal reasoning (Wang et al., 2024), commonsense reasoning (Zellers et al., 2019), and more. Advancing reasoning may explicitly or implicitly translate to improvements in several downstream NLP applications.

**Temporal reasoning.** Temporal reasoning has recently gained substantial attention (*e.g.*, Vashishtha et al., 2020; Nylund et al., 2023; Hu et al., 2023; Gurnee & Tegmark, 2023; Liu et al., 2023; Xiong et al., 2024; Beniwal et al., 2024; Jia et al., 2024). Much research focuses on enhancing LLMs' understanding of temporal concepts, primarily through pre-training and fine-tuning strategies to improve their temporal reasoning capabilities (*e.g.*, Ning et al., 2019; Zhou et al., 2020; Yang et al., 2023; Xiong et al., 2024; Jia et al., 2024).

Benchmark creation is another active area, with many benchmarks centered on knowledge graphs (*e.g.*, Jia et al., 2018; Neelam et al., 2021; Jia et al., 2021; Wang & Zhao, 2023; Chu et al., 2023; Su et al., 2024). While TempTabQA (Gupta et al., 2023) offers crowd-sourced questions based on Wikipedia infoboxes, the process is resource-intensive and prone to issues like LLM overuse by workers. The questions in Wang & Zhao (2023) are all multiple-choice, and do not require reasoning through a many temporal facts from a knowledge graph. The questions in Chu et al. (2023) are collected from ten existing real-world datasets, one of which requires reasoning through temporal facts provided in the context. In contrast, ToT goes beyond such datasets by providing controllable, comprehensive temporal relationship collections via synthetic graph generation. The questions in Timo Su et al. (2024) are grouped into two categories: math-time and pure-time. ToT-Artithmetic covers more domains in the math-time category and more focus on reasoning in the pure-time category. Xiong et al. (2024) recently proposed TGQA, a data set derived from the YAGO11k knowledge graph (Dasgupta

Table 1: Comparison of `ToT` against related benchmarks.

| Benchmark | Semantics | Arithmetic | Real-World | Synthetic | Hermetic | Implicit |
|---|---|---|---|---|---|---|
| TimeSensitiveQA (Chen et al., 2021) | ✓ | ✗ | ✓ | ✗ | ✗ | ✗ |
| StreamingQA (Liska et al., 2022) | ✓ | ✗ | ✓ | ✗ | ✗ | ✗ |
| TempLama (Dhingra et al., 2022) | ✓ | ✗ | ✓ | ✗ | ✗ | ✗ |
| TEMPTABQA (Gupta et al., 2023) | ✓ | ✗ | ✓ | ✗ | ✗ | ✓ |
| TEMPREASON (Tan et al., 2023) | ✓ | ✓ | ✓ | ✗ | ✗ | ✓ |
| TIQ (Jia et al., 2024) | ✓ | ✗ | ✓ | ✗ | ✗ | ✓ |
| TempUN (Beniwal et al., 2024) | ✓ | ✗ | ✓ | ✗ | ✗ | ✗ |
| TGQA (Xiong et al., 2024) | ✓ | ✗ | ✓ | ✗ | ✗ | ✗ |
| `ToT` (ours) | ✓ | ✓ | ✓ | ✓ | ✓ | ✓ |

et al., 2018). To prevent data leakage, TGQA changes each entity name to a name generated by GPT3.5 that is guaranteed to (i) align with the entity's type and (ii) not be otherwise present in YAGO11k. This strategy has several weaknesses. First, it can introduce spurious entity name correlations (LLMs could even potentially guess the original entities due to their adjacent relations). Second, it can generate factually incorrect or anti-commonsensical claims, for instance, if an entity's generated replacement name is a *real* name that is nonetheless not in YAGO11k. Finally, relying on GPT for copying facts introduces the potential for hallucinations to contaminate the dataset.

**Synthetic datasets.** A new trend in probing various LLMs capabilities, especially in the case of reasoning, is through synthetic data which allows for a more systematic evaluation. Previous work has developed synthetic datasets for probing reasoning capabilities including logical reasoning (Tafjord et al., 2021; Kazemi et al., 2023c; Saparov et al., 2023) and mathematical reasoning (Kazemi et al., 2023a; Srivastava et al., 2024). Most similar to our work, Fatemi et al. (2024) develop a synthetic probe for measuring the graph-based reasoning abilities of LLMs (Sanford et al., 2024; Perozzi et al., 2024). Our work extends this concept to the case of temporal reasoning with graph-like facts.

**Present work.** In this work, we introduce `ToT`, a novel benchmark for temporal reasoning generated synthetically. Unlike many existing benchmarks that rely on knowledge graphs, `ToT` aims to encompass a wider variety of graph structures. Our synthetic generation approach offers precise control over the type of data produced. Importantly, when evaluating LLMs against `ToT`, they cannot exploit their latent knowledge for shortcuts; instead, they must genuinely reason with the presented facts. This design promotes a more rigorous assessment of temporal reasoning capabilities in LLMs. Table 1 provides a comprehensive comparison of `ToT` with existing benchmarks across six key dimensions: **1- Semantics:** whether the benchmark has semantic-type questions, **2- Arithmetic:** whether the benchmark has arithmetic-type questions, **3- Real-world:** whether the benchmark has questions generated from real-world data, **4- Synthetic:** whether the benchmark has questions generated from synthetic data, **5- Hermetic:** whether the benchmark is sealed off from potential LLM training data, and **6- Implicit:** whether the benchmark includes implicit questions. Our analysis reveals that `ToT` is unique in incorporating all these question types while effectively mitigating training data leakage. Notably, TEMPREASON (Tan et al., 2023) only covers one category of the arithmetic operations as defined in Section 3.2.

## 3 `ToT`: A BENCHMARK FOR EVALUATING LLMS ON TEMPORAL REASONING

We propose that effective temporal reasoning hinges on two distinct skills: understanding the semantics and logic of time, and performing accurate temporal arithmetic. To measure and improve model performance along these independent axes, we create a dedicated task for each skill. By decoupling the evaluation of temporal semantics from arithmetic, we aim to provide a more nuanced analysis of LLM capabilities, pinpointing strengths and weaknesses in each aspect. Experiments on these tasks enable us to independently benchmark LLM performance on both dimensions.

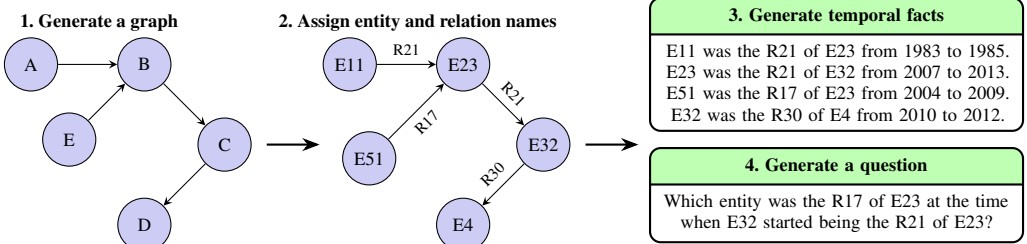

Figure 2: Steps for creating the `ToT-Semantic` dataset.

## 3.1 ToT-Semantic: A Synthetic Dataset

The first task we introduce, `ToT-Semantic`, consists of synthetic problems designed to highlight temporal semantics and logic in reasoning. This task is unique because it allows us to freely experiment with a wide range of temporal dependencies and manipulate the difficulty of the reasoning problem. This allows us to isolate and analyze the core reasoning capabilities of an LLM, separating them from any reliance on pre-existing parametric knowledge. To create the `ToT-Semantic` task, we take the following steps (summarized in Figure 2):

**Step 1: Generate a (random) graph.** We begin by generating random structures that we will then use to create temporal questions. To ensure we generate a diverse set of random structures for this purpose, we turn to the literature on graph structure generation. From it, we employ several existing algorithms for generating graphs of varying properties. This includes Erdős-Rényi (ER) graphs (Erdős & Rényi, 1959), scale-free networks (SFN) (Barabási & Albert, 1999), graphs following the Barabási–Albert (BA) model (Albert & Barabási, 2002) and stochastic block model (SBM) (Holland et al., 1983), as well as star and complete graphs. Each of these graph generation algorithms exhibits different properties and correspond to graphs that appear in different applications. For instance, Erdős-Rényi graphs are typically sparse with low average degree, while Barabási-Albert graphs are dense and exhibit power-law degree distributions. We leverage the NetworkX library for generating our random graphs. Additionally, we extracted anonymized EgoNets from Wiki-Data (Vrandečić & Krötzsch, 2014) by replacing the entity and relation names with generic names. We refer to this structure as *Anonymized Wikidata Extract (AWE)* in our experiments. We generate graphs with the number of nodes selected uniformly at random from the [5-30] interval. More details on the random graph generators used (with visualizations) are available in Appendix A.

**Step 2: Assigning entity and relation names.** Once we have an initial graph structure, we assign names to the nodes and relations to the edges. For each graph, we first decide a number of relation types to be assigned to the edges, and assign each of these relation types to one of one-to-one, one-to-many, many-to-one and many-to-many. Then, for each edge in the graph, we randomly assign between 1 to p (=3 in our experiments) relations types without violating the relation type arity.

**Step 3: Generate temporal facts.** Then, for each edge $(u, v)$ labeled with a relation $r$, we assign a valid time interval $[t1, t2]$ that respects the relation types, and turn the tuple $(u, v, r, t1, t2)$ into a textual temporal fact using a template.

**Step 4: Question generation.** Having generated the random graphs, we then create questions about those graphs. We consider eight types of questions that are frequently used in day-to-day life and are common in various benchmarks. **EventAtTimeT**: asking which entity had some relation R with entity E at some T; **EventAtWhatTime**: asking at what time a relation R between two entities E1 and E2 started/ended; **NumEventsInTimeInterval**: asking how many entities had relation R with entity E between T1 to T2; **BeforeAfter**: asking which entity had relation R with E1 right before/after E2; **EventAtTimeOfAnotherEvent**: asking when E1 had relation R1 with E2, which entity had relation R2 with E3; **FirstLast**: asking which entity was the first to have relation R with E; **RelationDuration**: Asking the k-th time relation R happened between E1 and E2, how long did it last; and **timeline**: Asking to sort the entities that had relation R with E chronologically.

To create any of the above questions, we keep sampling graphs and fact(s) from the graph until a proper question of the desired type can be created for that graph and for that fact. For example, to

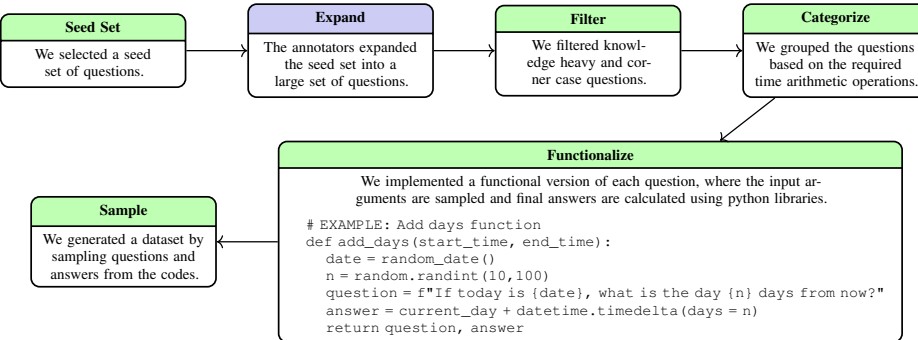

Figure 3: Steps for creating the `ToT-Arithmetic` dataset. The green and blue colors represent the operations done by the authors and the annotators respectively.

create a *BeforeAfter* question, we keep sampling a graph $G$ and fact $F = (S, R, O, T1, T2) \in G$ until we have a case where there is a unique entity $E$ that was the $R$ of $O$ right before $[T1, T2]$.

Following the above two steps, we generated 10 questions per graph generation and per question type. We sorted the facts in five different ways as will be discussed later. This gives as a benchmark with a total of $7 * 8 * 5 * 10 = 2800$ questions, where 7 is the number of graph generation algorithms, 8 is the number of question types, 5 is the number of sorting algorithms, and 10 is the number of samples we generated. Example questions of each category type are shown in Table 2.

## 3.2 ToT-ARITHMETIC: A TEMPORAL ARITHMETIC DATASET

Our second task, `ToT-Arithmetic`, shifts from synthetic data to a real-world focus. This task moves beyond understanding the logic and semantics of time and delves into the practical application of mathematical operations within a temporal context. Through it, we are able to measure an LLM's proficiency in temporal arithmetic and its practical utility in handling time-related computations.

To create a large time-arithmetic dataset that covers a wide variety of problems, we took the steps summarized in Figure 3. We explain each step in more detail below.

- **Seed Set:** By examining the existing benchmarks and the kind of temporal arithmetic questions that arise in them and through searching the web, we gathered a small set of initial questions.

- **Expand:** We presented our seed set to 15 annotators who were tasked to propose either new time arithmetic questions that were not in our seed set, or to provide questions corresponding to other scenarios or question templates where one requires to do similar time arithmetic operations to one of the questions in our seed set. We gathered a large list of questions through this process.

Table 2: Example for each question type in the `ToT-Semantic` dataset.

| Question Type | Example |
|---|---|
| EventAtTimeT | Find the entity that evidently was the R17 of E69 in year 1932. |
| EventAtWhatTime | At what time did E69 start being the R90 of E22? |
| NumEventsInTimeInterval | Find the number of unique entities that were the R82 of E27 between 1952 to 1957. Relations that ended in 1952 or started in 1957 must be counted. |
| BeforeAfter | Immediately before E59, which entity was the R20 of E6? |
| EventAtTimeOfAnotherEvent | E94 was the R82 of which entity at the time when E83 started being the R20 of E59? |
| FirstLast | Which entity was the first that was the R35 of E91? |
| RelationDuration | When E24 was the R53 of E11 for the 2nd time, for how many years did the relation last? The duration can be computed by subtracting the start time from the end time. |
| Timeline | Which entities were the R17 of E69? |

- **Filter:** We manually went through all the questions and filtered the ones that were focusing on corner cases, or that required extensive knowledge (*e.g.*, requiring to memorize the entire calendar).

- **Categorize:** We then grouped the remaining problems into seven categories, shown with examples in Table 3. Categories are formed based on the time arithmetic operations required, as follows: **AddSubtract**: adding or subtracting a number (corresponding to days, weeks, minutes, hours, etc.) from a date or time; **Compare**: comparing a number of dates/times provided in different formats chronologically; **Duration**: computing the difference between two dates/times; **Schedule**: finding mutual free spots within multiple blocked times; **Timezone**: involving dealing with different timezones; **Trick**: involving questions with slight twists; and **MultiOp**: involving questions where multiple of the above operations are needed.

- **Funcionalizing:** Following (Srivastava et al., 2024), we implemented a functional version of each problem to enable sampling different values for each question and solving based on those values. A functional version of one of our simple problems is provided in Figure 3.

- **Sampling:** We then sampled questions and answers from our functionalized problems, proportional to the number of different problems that fell under each category. Specifically, we sampled 350 for AddSubtract, 350 for Compare, 200 for Duration, 250 for Schedule, 100 for Timezone, 250 for Trick, and 350 for MultiOp. This resulted in a dataset with 1850 questions in total.

## 3.3 QUALITY CHECK

For both tasks, we did multiple rounds of quality checks where we verified 1) whether the generated labels are correct, and 2) whether the question is clear and the provided instructions are sufficient to know in what format the output should be produced. This procedure was done until no more issues could be found in the dataset.

Table 3: Examples for each question type in the `ToT-Arithmetic` dataset.

| Category | Example |
|---|---|
| AddSubtract | Your driver's license expires on 18 May, 2017. You receive a renewal notice saying it can be renewed 117 days in advance. What's the earliest date you can renew your license? |
| Compare | E42 was discovered in 14 April, 52 BC and E11 was discovered in 05 October, 530 BC. Which was discovered earlier? |
| Duration | Stella and William were born on 1999-Dec-16 and 2000-Oct-03 respectively. When William was 400 days old, how old was Stella in days? |
| Schedule | Lucas is available from 11 to noon and also from 3:30 to 5. Asher is available from 11 to 12:30 and also from 4 to 5. They want to have a 30 minute meeting. The meeting has to start on the hour or half hour. How many possibilities are there for the meeting time? |
| Timezone | Flight departs location A at 11:08 (24hr) UTC(+0000). It reaches location B at 07:23:20 PM IST(+0530). What is the total time duration taken to fly? |
| Trick | If the date for the day before tomorrow in yyyy-mm-dd format is 2016-01-20, what is the date 27 days from now in the same format? |
| MultiOp | Alex solves 2 puzzles in 4 hours, 50 minutes, and 22 seconds. What is the time taken by them to solve 6 puzzles, at the same pace. |

## 4 EXPERIMENTS AND RESULTS

In this study, we evaluate the performance of five frontier large language models (LLMs) on our benchmark. The models evaluated include Claude-3-Sonnet (Anthropic, 2024), Mistral Large (2407) (Team, 2024), GPT-4 (Achiam et al., 2023), Gemini 1.5 Pro (Reid et al., 2024), and GPT-4o (OpenAI, 2024). For GPT-4, we employed GPT-4 Turbo for the `ToT-Semantic` task, as it supports a larger context size, and standard GPT-4 for the `ToT-Arithmetic` task due to its superior performance. The same variant of GPT-4o was used for both tasks.[1]

---

[1]The experiments were done in Summer 2024 with the latest versions of the models at the time.

Table 4: LLM accuracy on temporal reasoning tasks by graph structure.

| Graph | Claude-3-Sonnet | Mistral Large | GPT-4 | Gemini 1.5 Pro | GPT-4o | Average |
|---|---|---|---|---|---|---|
| BA | 48.50 | 63.00 | 63.25 | 62.75 | **72.00** | 61.90 |
| Complete | 34.00 | 32.75 | 40.25 | **52.50** | 51.00 | 42.10 |
| ER | 42.25 | 42.25 | 48.75 | 60.50 | **62.25** | 51.20 |
| SBM | 42.00 | 48.50 | 50.75 | 57.75 | **61.75** | 52.15 |
| SFN | 58.75 | 77.75 | 75.25 | 75.75 | **86.00** | 74.70 |
| Star | 59.50 | 77.50 | 80.25 | 74.25 | **81.75** | 74.65 |
| AWE | 68.75 | 88.50 | 92.00 | 87.50 | **94.00** | **86.15** |
| **Average Rank** | 4.75 | 3.50 | 2.75 | 1.43 | **1.12** | |

In our experiments, we aim to answer the following questions:

**RQ1:** What is the effect of the temporal structure on the LLM performance?
**RQ2:** What kind of temporal questions are easier/harder for LLMs to answer?
**RQ3:** How important is the order of the facts and what is the best way of ordering them?
**RQ4:** How well do the frontier models perform on time semantics and time arithmetic?

## 4.1 INVESTIGATING THE IMPACT OF TEMPORAL STRUCTURE ON LLM TEMPORAL REASONING

In different applications where temporal reasoning arises, the structure of the facts can be different. Some tasks may provide all the information about an entity (corresponding to a star graph) and ask questions about it, whereas in some others, such as social networks, the structure of the facts may follow a power-law distribution. We study whether the inherent temporal structure of a problem might influence an LLM's ability to reason over its data. Drawing inspiration from recent work analyzing graph neural networks (Palowitch et al., 2022; Tsitsulin et al., 2022; Yasir et al., 2023; Fatemi et al., 2024), this section aims to quantify how different temporal dependencies affect an LLM's temporal reasoning capabilities using graph generators to create many different kinds of temporal structure.

The graph structure of the temporal relationships significantly affects LLM performance, as demonstrated in Table 4. Notably, GPT-4 accuracy more than doubled between complete graphs (40.25%) and AWE graphs (92.00%). Also, Mistral Large accuracy varied drastically across graph types, from 32.75% for complete graphs to 88.50% for AWE graphs. This highlights a critical gap in temporal reasoning research, which has largely overlooked the diverse graph structures and reasoning tasks found in real-world applications, instead focusing primarily on specific knowledge graphs (like YAGO11k). This may explain the superior performance of LLMs on AWE graphs in our experiments, with GPT-4o nearly solving the task with 94.00% accuracy.

### 4.1.1 INFLUENCE OF GRAPH SIZE ON LLM PERFORMANCE

A key question is how different models behave as a function of the size of a graph, measured in terms of the number of edges (facts) and nodes (entities). As illustrated in Figure 4, increasing either the number of edges or nodes in the `ToT-Semantic` dataset mostly leads to a decrease in LLM performance. We observe, however, that different models are affected differently. For example, for the smaller graphs with $< 250$ edges, GPT-4o outperforms the other models, whereas when the size increases to $> 1000$ edges, Gemini 1.5 Pro outperforms the other models. Moreover, we observe that while the performance of GPT-4o and Gemini 1.5 Pro does not degrade much after a certain point of increasing the number of edges (specifically, for the last three buckets), other models' performances keep decreasing (with the exception of GPT-4 at the last bucket).

Table 5: Average number of nodes and edges by graph structure.

| Graph | #nodes | #edges |
|---|---|---|
| BA | 17.41 | 144.07 |
| Complete | 17.25 | 619.86 |
| ER | 16.18 | 316.4 |
| SBM | 17.51 | 368.15 |
| SFN | 17.52 | 53.46 |
| Star | 16.16 | 34.12 |
| AWE | 18.99 | 25.41 |
| **Average** | 17.29 | 223.07 |

The above results raise the question of whether the graph structure's impact observed in Section 4.1 is merely a consequence of varying graph sizes. To address this, we present the average number of nodes and edges for each graph structure in Table 5. While the average number of nodes does not appear to consistently influence LLM performance across structures, the number of edges does show some correlation. However, there are exceptions. For instance, SBM graphs have far more edges on

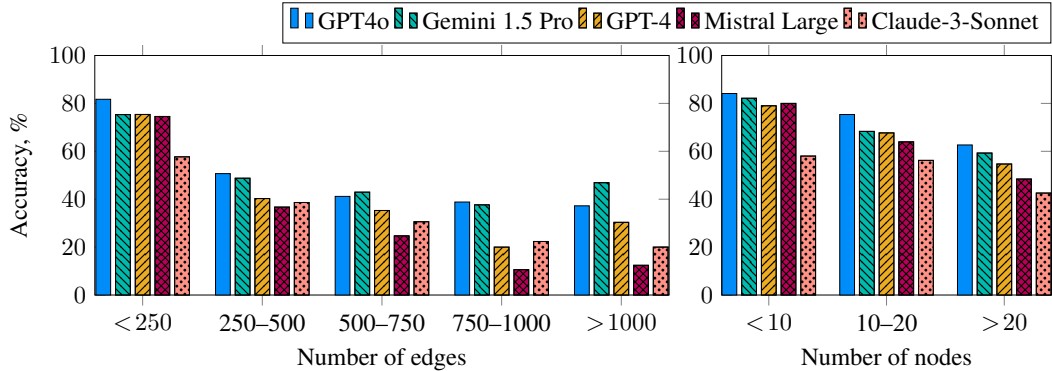

Figure 4: Accuracy of models for different number of edges and nodes.

overage than ER graphs, yet the average performance of models across ER graphs is lower than SBM graphs. Also, SFN graphs have far more edges on average than Star graphs, yet GPT-4o performs better on SFN graphs than Star graphs. This indicates that both the number of edges and the specific structure of the graph play a significant role in determining LLM performance. As for number of nodes, AWE graphs have more nodes on average compared to the other graph structures, yet the average performance of models across AWE is the highest across all (see Table 4).

## 4.2 EFFECTS OF TEMPORAL QUESTION TYPE ON LLM TEMPORAL REASONING

Table 6: LLM accuracy on temporal reasoning by question category.

| Temporal Question Type | Claude-3-Sonnet | Mistral Large | GPT-4 | Gemini 1.5 Pro | GPT-4o | Average |
|---|---|---|---|---|---|---|
| EventAtTimeT | 47.14 | 64.86 | 65.43 | **72.29** | 71.43 | 64.23 |
| EventAtWhatTime | 90.29 | 90.00 | 89.43 | 93.14 | **96.86** | **91.94** |
| NumEventsInTimeInterval | 29.71 | 57.14 | 61.43 | 59.14 | **63.71** | 54.23 |
| BeforeAfter | 53.14 | 56.57 | 55.43 | 52.86 | **64.00** | 56.40 |
| EventAtTimeOfAnotherEvent | 50.00 | 57.43 | 67.14 | 71.43 | **75.71** | 64.34 |
| FirstLast | 68.57 | 57.71 | 67.71 | 68.57 | **73.71** | 67.25 |
| RelationDuration | 41.43 | 76.57 | 80.00 | 84.57 | **88.86** | 74.29 |
| Timeline | 24.00 | 31.14 | 28.29 | 36.29 | **38.57** | 31.66 |
| **Average Rank** | 4.31 | 3.75 | 3.37 | 2.44 | **1.12** | |

In this experiment, we systematically investigate the impact of different temporal tasks on the reasoning ability of LLMs. We quantify this impact by evaluating model performance across a variety of tasks, as summarized in Table 6.

**Task type and reasoning requirements.** A key question in our investigation is whether the type of temporal task and the associated reasoning requirements influence LLM performance. The `ToT-Semantic` dataset includes questions of varying difficulty levels, which can be categorized based on the reasoning type: **Single-fact solutions:** Questions EventAtTimeT and EventAtWhatTime require retrieving one single fact and answering the question directly based on the retrieved fact. **Multi-fact solutions:** The remaining questions require retrieving multiple facts and performing operations (*e.g.*, counting, sorting) to extract relevant information and formulate an answer.

LLMs consistently demonstrate superior performance on tasks requiring the retrieval of a single fact compared to those necessitating the integration of multiple facts. This performance gap can be attributed to the increased cognitive demands associated with multi-fact tasks. While single-fact questions primarily rely on the identification and extraction of relevant information, multi-fact questions demand a deeper comprehension and synthesis of retrieved information.

**Performance variations within question types.** Even within zero-order reasoning tasks, LLMs demonstrate varying levels of proficiency. For example, EventAtTimeT and EventAtWhatTime are structurally similar, yet LLMs tend to excel at the latter. We hypothesize that this performance

difference may be attributed to the fact that EventAtTimeT requires a simple time arithmetic operation to recognize that a timestamp $T$ falls within a time interval $[T1, T2]$, whereas EventAtWhatTime does not require any time arithmetic operation.

**Analysis on Timeline questions.** Timeline questions are the most difficult category of questions for the models according to Table 6. An analysis of these questions reveals that they pose the greatest challenge across all tasks. To answer these questions, typically structured as "Sort the entities that were the R17 of E69 chronologically?", the model needs to extract multiple entities (in the `ToT-Semantic` dataset, every

Table 7: Precision and recall on timeline questions.

| Graph structure | All | | Complete | |
|---|---|---|---|---|
| | Precision | Recall | Precision | Recall |
| Claude-3-Sonnet | 0.73 | 0.75 | 0.56 | 0.54 |
| Mistral Large | 0.62 | 0.65 | 0.30 | 0.33 |
| GPT-4 | 0.60 | 0.56 | 0.36 | 0.23 |
| Gemini 1.5 Pro | **0.81** | **0.83** | **0.82** | **0.65** |
| GPT-4o | 0.78 | 0.74 | 0.69 | 0.51 |

timeline question has more than one entity in its label), and then do temporal arithmetic to sort them temporally. To further analyze the models on these questions, we calculated the average precision and recall for each model in Table 7, where precision shows what percentage of the entities extracted by the model are correct entities (i.e. must be included in the timeline) and recall shows what percentage of the correct entities have been extracted by the model. We report the results once averaged over all graph structures and once only for complete graphs (the most challenging graph structure). Gemini 1.5 Pro demonstrates superior precision and recall, aligning with its relatively high accuracy observed in Table 6. The only model outperforming Gemini 1.5 Pro on timeline questions is GPT-4o. The fact that the precision and recall of GPT-4o is lower than that of Gemini but its overall performance on timeline questions is higher shows that Gemini is better at retrieving the correct entities but worse at arithmetic operation (as also confirmed later in Section 4.4). Moreover, GPT-4, despite having higher accuracy than Claude-3-Sonnet on timeline questions, exhibits the lowest precision and recall. This suggests that GPT-4 frequently outputs fewer entities than are present in the true answers ($50\%$ of the times), leading to missed correct entities (lower recall) and a higher proportion of false positives among its predictions (lower precision).

Since complete graphs pose the greatest difficulty among all graph structures (Table 4), we provide a separate analysis of average precision and recall for these graphs in the final two columns of Table 7. Notably, all models except Gemini 1.5 Pro experienced declines in both precision and recall on complete graphs, whereas Gemini was primarily impacted in terms of recall.

Table 8: LLM accuracy on temporal reasoning tasks as a function of the order of the facts.

| Order of facts | Claude-3-Sonnet | Mistral Large | GPT-4 | Gemini 1.5 Pro | GPT-4o | Average |
|---|---|---|---|---|---|---|
| Shuffle | 45.71 | 55.71 | 60.71 | 63.04 | **68.93** | 58.82 |
| RelationAndStartTime | 54.29 | 63.93 | 65.36 | 68.57 | **72.14** | 64.86 |
| StartTimeAndRelation | 47.68 | 59.11 | 60.54 | 64.64 | **65.89** | 59.57 |
| StartTimeAndTarget | 49.11 | 60.89 | 61.61 | 65.18 | **70.00** | 61.36 |
| TargetAndStartTime | 73.57 | 67.50 | 62.60 | 75.00 | **81.07** | **71.95** |

## 4.3 IMPACT OF TEMPORAL FACT ORDER ON LLM PERFORMANCE

A noteworthy question arises regarding the potential influence of fact order on LLM performance in temporal reasoning tasks. To investigate this, we conducted experiments on `ToT-Semantic` dataset. We sorted the facts using different methods: **Shuffle:** randomizing the order of facts; **RelationAndStartTime:** prioritizing facts based on their relation name, then by start time; **StartTimeAndRelation:** prioritizing facts based on start time, then by relation name; **StartTimeAndTarget:** prioritizing facts based on start time, then by the target entity; **TargetAndStartTime:** Prioritizing facts based on the target entity, then by start time.

Ideally, LLMs should exhibit robustness to the order in which facts are presented, given the independent nature of each fact. However, as shown in Table 8, our observations reveal a significant impact of fact order on LLM performance. Notably, performance is consistently lowest when facts are presented in a shuffled order and consistently highest when facts are sorted based on the target entity and start time (TargetAndStartTime). We also observe that some sorting strategies such as StartTimeAndRelation are only slightly better than the shuffled order, thus revealing that not any kind of ordering is ideal for LLMs. This finding offers valuable practical insights into how facts should be structured when temporal reasoning is a key component of the LLM task. By organizing

facts in a manner that aligns with the temporal flow of the narrative or task, we can potentially enhance LLM performance and ensure more accurate and reliable results. While previous work has shown that ordering premises in the correct order of chain-of-thought solution improves LLM's logical reasoning (Chen et al., 2024; Saparov & He, 2022), our results extend that to general-purpose temporal orderings (independent of the chain-of-thought).

Table 9: LLM accuracy on the `ToT-Arithmetic` dataset by question type.

| Category | Claude-3-Sonnet | Mistral Large | GPT-4 | Gemini 1.5 Pro | GPT-4o | Average |
|---|---|---|---|---|---|---|
| AddSubtract | 58.57 | 61.14 | 76.28 | 71.14 | **76.29** | 68.68 |
| Compare | 39.14 | 62.29 | 63.14 | 55.43 | **66.57** | 57.30 |
| Duration | 15.00 | **17.50** | 16.00 | 13.50 | 15.00 | 15.40 |
| Schedule | 29.60 | 44.40 | 43.60 | 40.00 | **53.20** | 42.16 |
| Timezone | 74.00 | 87.00 | 88.00 | 90.00 | **92.00** | 86.20 |
| Trick | 40.40 | 44.80 | 45.60 | 41.20 | **53.20** | 45.04 |
| MultiOp | 26.57 | 54.86 | 46.86 | **62.57** | 46.86 | 47.54 |
| Average Rank | 4.71 | 2.71 | 2.43 | 3.28 | **1.57** | |

## 4.4 Temporal semantics vs temporal arithmetic

This study examined the performance of temporal arithmetic capabilities in LLMs using the `ToT-Arithmetic` dataset. Results, as shown in Table 9, indicate that the models consistently excelled in Timezone questions, while struggling the most with Duration questions. This superior performance in Timezone questions could be attributed to the abundance of information about various timezones available online, compared to other question types. Scheduling and Trick questions also proved challenging for LLMs, likely due to their creative nature and requirement for deeper reasoning. In contrast, AddSubtract results were relatively strong, potentially reflecting LLMs' optimization for mathematical reasoning and their ability to apply that knowledge to temporal reasoning tasks.

**Analysis on Duration questions.** Analysis of Duration questions in the `ToT-Arithmetic` dataset revealed them to be the most challenging for the evaluated models. Notably, the most common error among incorrect answers was a deviation of precisely one day from the ground truth label. Specifically, when GPT-4 or Gemini 1.5 Pro erred on Duration questions, approximately 21% and 25% of its responses were within one day of the ground truth, respectively. This suggests that LLMs can approximate the correct calculation but often stumble in the final steps, highlighting a gap in their ability to execute complex arithmetic with precision.

**Common failure: direction.** One frequent error in `ToT-Arithmetic` occurs when determining the number of months between two dates. For example, from February 11th, 2002, to October 11th, 2002, the correct duration is eight months, but the model sometimes incorrectly calculates it as four months. This issue is particularly noticeable in questions that involve going back in time, such as: "Sam's birthdate is October 11th, 1996. Today is February 25th, 2002. Calculate Sam's age in days."

**Common failure: leap year calculation.** Another frequent error in `ToT-Arithmetic` arises when determining the number of days between two dates that span multiple years. Incorrectly accounting for leap years, which have an extra day (February 29th), often leads to inaccurate results.

## 5 Conclusion

In conclusion, we introduce Test of Time (`ToT`), a novel benchmark designed to assess LLMs' temporal reasoning abilities in a more comprehensive and controlled manner than existing work. Our two-pronged approach, encompassing both semantic and arithmetic tasks, enables a nuanced evaluation of temporal reasoning. Through extensive experiments with `ToT`, we have gained valuable insights into the strengths and weaknesses of current LLMs in these critical aspects of temporal reasoning. By open-sourcing our datasets and evaluation framework, we hope to stimulate further research and development in this field, ultimately contributing to the advancement of LLM capabilities in complex reasoning tasks.

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

## A  DESCRIPTION OF GRAPH GENERATORS.

Here we detail each graph generator used to create the examples in ToT. We note that every collection of temporal facts, where each fact is a relationship between two entities, can be expressed as a temporal graph with nodes as entities. ToT specifically targets LLM reasoning ability over such collections. We do not claim that graph generators are the only way to construct such a benchmark. However, because all temporal fact collections contain an underlying graph, we propose a generation framework based on graph models to produce benchmark examples. We argue that a framework that exposes generation of the static graph backbone is more controllable and allows for a benchmark that is more comprehensive w.r.t. the variety and complexity of temporal relationships between generated entities.

First, we cover the six *random* graph generators used to create the synthetic examples. All random graph generators are probabalistic models which take hyperparameters that control the expected macro-properties of each graph (Palowitch et al., 2022):

- Erdős-Rényi (ER) (Erdős & Rényi, 1959): This model takes an edge probability parameter $p$ and generates each edge with probability $p$, i.i.d. over all possible edges.

- Scale-Free Networks (SFN) (Barabási & Albert, 1999): a graph is grown by a sequence of steps, each step either (1) adding a new node and connecting it to an existing node, or (2) adding an edge between two existing nodes. Input parameters control the probability of these events. This process generates a "scale-free" power law of node degrees, in sharp contrast to the ER model.

- Barabási–Albert (BA) model (Albert & Barabási, 2002): a graph is grown by a sequence of steps, each step adding a new node to the graph, and connecting the node to $m$ existing nodes with probability proportional to their current degree. Similar to SFN, this process also generates a "scale-free" graph with a particular distribution known as the Barabási–Albert model.

- Stochastic Block Model (SBM) (Holland et al., 1983): This graph model can be thought of as clustered ER. It divides $n$ nodes into $k$ clusters, and then connects two nodes with probability $p$ if they are in the same cluster, else with probability $q$ if they are in different clusters. $k$, $p$, and $q$ are all hyperparameters.

- A star-graph generator creates a "star" graph on $n$ nodes: node 0 is the center of the star, and all other nodes connect to it (and only it).

- A complete-graph generate creates a "complete" graph on $n$ nodes, in which all nodes are connect to each other node.

An example from each of the above graph generators is shown in Figure 5. In the figure, edges are annotated with temporal relationships in the format `relation_id: [interval_1, ..., interval_k]`. Note that each edge can have multiple relationships, and each relationship can have multiple intervals. The visualization shows the diversity of temporal knowledge graphs that our framework is able to generate. We note that while our study was limited to parametric graph generators in this work, the field of graph machine learning (Chami et al., 2022) offers many options for both modeling (Perozzi et al., 2014) and learning (Halcrow et al., 2020; Fatemi et al., 2021; Rozemberczki et al., 2021; Fatemi et al., 2023) link structure.

Second, we describe our Anonymized Wikidata Extract (AWE) strategy for creating anonymized questions from real-world data. We first identify the 78 most common relations in WikiData that specify time-bound entity relationships. Each relation encodes a temporal edge between two entities. To match the schema of our synthetic graphs, we convert each time specification on each edge to an interval. Then, for each entity in the graph, we extract the *ego-graph* of the entity by (1) collecting the entity and all its neighbors and (2) collecting all edges (along with temporal information) between nodes collected in (1). This process produces a temporal graph with a schema identical to those produced from random graph generators. Before generating questions from the graphs, we anonymize them by (a) mapping each entity name to a unique identifier such as `E679`; and then (b) mapping each relation name to A unique identifier such as `R3`. We then generate questions from the graph as described in 3.1.

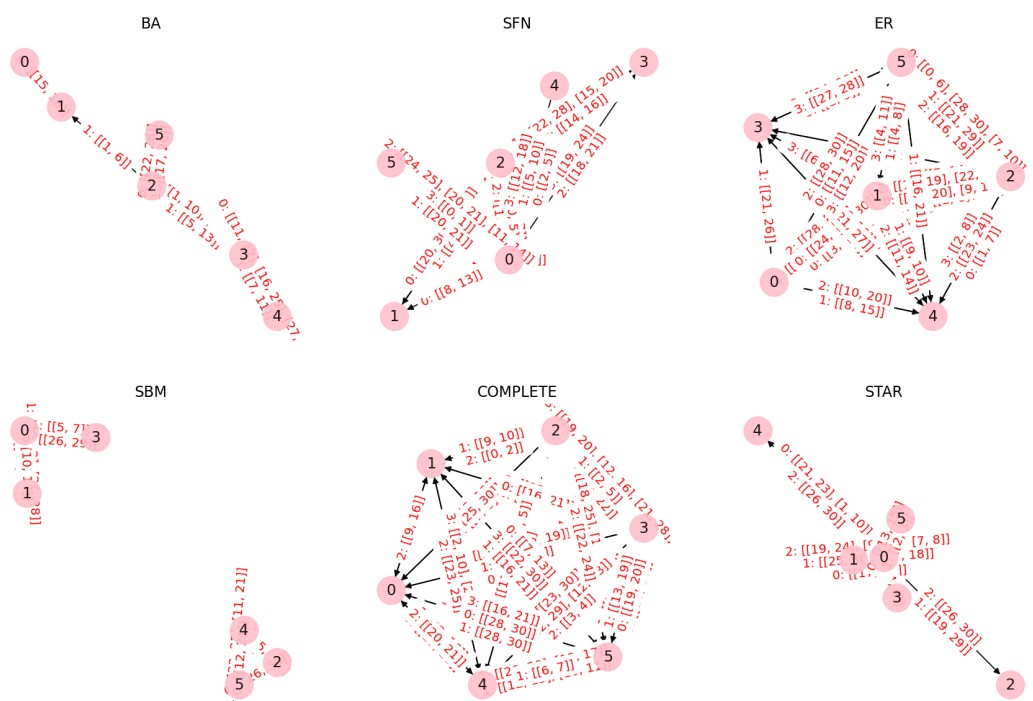

Figure 5: A visualization of a representative graph from each graph generator: Erdős-Rényi (ER), Scale-Free Networks (SFN), Barabási–Albert (BA), Stochastic Block Model (SBM), star-graph, and complete-graph.

## B  DETAILS OF QUESTION GENERATION.

Given a graph with temporal facts, generating logically-consistent questions from our list of diverse question types (see Table 2) is non-trivial. To generate the total question set, we loop through generated graphs, choose a question type uniformly-at-random, and then attempt to fill the question type template with facts from the graph. The exact algorithmic procedure is given below. Note that the SAMPLEFACTS routine will vary significantly depending on the question type. For some questions, it is sufficient to generate a single fact and check if the question can be generated. For other question, multiple facts must be sampled (sometimes sequentially, in a BFS fashion) and checked for cohesion with the template. We do all of this in a brute-force manner.

---

**Algorithm 1** Generate all questions from a certain question type `template`.

---

1: **Procedure** GENERATEQUESTIONS($\mathcal{G}$, $n$, `template`, $m$)
2: $\mathcal{Q} \leftarrow \phi$
3: **for** $i \in [n]$ **do**
4:     $G \leftarrow$ SAMPLEGRAPH($\mathcal{G}$)
5:     $q \leftarrow$ GENERATEQUESTION($G$, `template`, $m$)
6:     **if** $q = \phi$ **then**
7:         continue
8:     **end if**
9:     $\mathcal{Q} \leftarrow \mathcal{Q} \cup \{q\}$
10: **end for**
11: **return**  $\mathcal{Q}$

---

---

**Algorithm 2** Generate a single question from a graph with maximum trials $m$.

---

 1: **Procedure** GENERATEQUESTION($G$, template, $m$)
 2:   $q = \phi$
 3:   **for** $j \in [m]$ **do**
 4:     $F \leftarrow$ SAMPLEFACTS($G$, template)
 5:     $q \leftarrow$ template($F$)
 6:     **if** $q \neq \phi$ **then**
 7:       **break**
 8:     **end if**
 9:   **end for**
10:   **return** $q$

---

## C   LARGE-SCALE ToT-Semantic EXPERIMENTS

To facilitate a more comprehensive analysis and enable deeper insights, we expanded our synthetic dataset significantly. This enlarged dataset now encompasses approximately $50,000$ examples, a substantial increase from the previous set of around $3,000$ examples. We anticipate that this expanded resource will prove valuable for future research endeavors that necessitate a larger and more diverse synthetic dataset. Due to the computational demands of evaluating all LLMs on this large dataset, results are reported solely for Gemini 1.5 Pro.

**Impact of Graph Structure on LLM Accuracy.**   Our initial experiment with this expanded dataset involved replicating the graph structure analysis. As illustrated in Table 10, graph structure continues to exert a significant influence on the final accuracy of the LLM, even within this larger dataset.

Table 10: LLM temporal reasoning by graph structure on the larger set of ToT-Semantic.

| Graph Structure | Accuracy (%) |
|---|---|
| BA | 70.96 |
| Complete | 51.07 |
| ER | 61.85 |
| SBM | 60.32 |
| SFN | 79.13 |
| Star | 73.77 |
| AWE | 88.72 |
| Average | 69.40 |

**Impact of graph structure and temporal task on LLM performance.**   Our second experiment examined the accuracy of the model across various question types and graph generators. The expanded dataset provided sufficient examples per category, enabling more robust results. The results are reported in Table 11. Consistent with our earlier findings, single-fact questions generally outperformed multi-fact questions. Notably, the highest accuracy was observed for EventAtWhatTime in single-fact questions and RelationDuration in multi-fact questions. This alignment with the results from the smaller dataset reinforces their significance and suggests that the smaller dataset serves as a reliable proxy for the larger one.

**Impact of Graph Structure and order of facts on LLM Performance.**   In this experiment, we evaluated LLM performance across various combinations of graph structure and fact order. The results, presented in Table Table 12, reveal that the target_and_start_time ordering consistently yields the best performance across the expanded dataset, regardless of graph structure. Conversely, the shuffle ordering consistently underperforms across most graph structures.

## D   EVALUATION PROCESS

We adopted a structured approach to ensure consistent evaluation. The LLM prompts incorporate specific guidelines for output formatting, requiring a JSON structure with fields like 'explanation'

Table 11: Impact of graph structure and question type on a larger set of `ToT-Semantic`.

| Temporal task | BA | Complete | ER | SBM | SFN | Star | AWE | Average Rank |
|---|---|---|---|---|---|---|---|---|
| EventAtTimeT | 74.46 | 54.22 | 65.54 | 68.07 | 80.84 | 76.75 | 91.93 | 3.57 |
| EventAtWhatTime | 98.19 | 81.69 | 90.72 | 90.48 | 98.31 | 98.43 | 97.95 | **1.00** |
| BeforeAfter | 53.49 | 34.46 | 48.07 | 45.66 | 68.55 | 58.80 | 73.98 | 7.00 |
| EventAtTimeOfAnotherEvent | 76.99 | 52.89 | 62.53 | 65.18 | 84.82 | 85.78 | 90.48 | 3.79 |
| FirstLast | 70.84 | 49.04 | 61.69 | 55.66 | 87.23 | 68.80 | 92.53 | 4.43 |
| NumEventsInTimeInterval | 57.71 | 40.84 | 54.22 | 49.64 | 64.22 | 70.84 | 83.73 | 6.14 |
| RelationDuration | 88.55 | 80.60 | 83.49 | 82.77 | 87.47 | 88.80 | 90.48 | **2.36** |
| Timeline | 47.47 | 14.82 | 28.55 | 25.06 | 61.57 | 41.93 | 88.67 | 7.71 |

and 'answer'. This standardized output facilitated automated evaluation through parsing the JSON, extracting the answer field(s), and comparing to the golden label. Here are examples of instructions in the prompt (please see below for the full prompt):

**Example from ToT-Semantic:** Answer the following question based on the temporal facts assuming the facts are unidirectional. Output only a valid JSON string with two fields: "explanation" and "answer". Do not output anything else. The explanation field contains your reasoning. The answer field contains the value corresponding to your final answer.

**Example from ToT-Arithmetic:** Return your answer as a JSON in the following format: JSON = "explanation": <your step by step solution>, "answer": "day_of_week".

This prompting method ensured clear instructions for the model's output format. Our experiments showed consistent adherence to these instructions, demonstrating the effectiveness of our prompt design and leading to a robust and straightforward evaluation process.

Table 12: Impact of graph structure and sorting type on a larger set of `ToT-Semantic`.

| Order of facts | BA | Complete | ER | SBM | SFN | Star | AWE | Rank Average |
|---|---|---|---|---|---|---|---|---|
| relation_and_start_time | 73.42 | 52.03 | 64.98 | 61.45 | 81.93 | 74.32 | 90.36 | 2.00 |
| shuffle | 66.72 | 44.65 | 54.74 | 54.14 | 74.17 | 72.74 | 85.02 | 4.71 |
| start_time_and_relation | 67.55 | 46.31 | 57.76 | 55.72 | 77.86 | 72.14 | 88.48 | 4.00 |
| start_time_and_target | 68.60 | 46.61 | 58.96 | 55.95 | 78.31 | 70.78 | 88.63 | 3.29 |
| target_and_start_time | **78.54** | **65.74** | **72.82** | **74.32** | **83.36** | **78.84** | **91.11** | **1.00** |

---

**Example from ToT-Semantic**

**Prompt:** Here is a set of temporal facts:
E92 was the R17 of E69 from 1980 to 1988.
E69 was the R17 of E24 from 1972 to 1981.
E59 was the R17 of E69 from 1999 to 2000.
E69 was the R90 of E42 from 1997 to 2001.
E11 was the R17 of E69 from 1977 to 1980.
E92 was the R17 of E69 from 1995 to 1999.
E69 was the R90 of E24 from 1977 to 1978.
E69 was the R90 of E42 from 1980 to 1982.
E69 was the R17 of E42 from 1986 to 1994.
E92 was the R90 of E69 from 1993 to 1996.
E69 was the R90 of E42 from 1974 to 1975.
E59 was the R90 of E69 from 1999 to 2000.
E69 was the R90 of E42 from 1983 to 1987.

Answer the following question based on the temporal facts assuming the facts are unidirectional. Output only a valid JSON string with two fields: "explanation" and "answer". Do not output anything else. The explanation field contains your reasoning. The answer field contains the value corresponding to your final answer.

**Question:** Find the entity that was the R90 of E69 immediately after E92 was the R90 of E69.

**Model response (Gemini 1.5 pro):** {"explanation": "E92 was the R90 of E69 from 1993 to 1996. The entity that was the R90 of E69 immediately after E92 was E59 from 1999 to 2000.", "answer": "E59" }

**Label:** E59

---

**Example from ToT-Arithmetic**

**Prompt:** If right now, it is 22:00 (24hr) in Location A (EST -0500), what time is it in Location B (PST -0800)? Answer with time and day in json. Eg: JSON = "explanation": <your step by step solution>, "day": "same_day/+x/-x", "time": "HH:MM:SS", where day=same_day if the answer is the same day, or +x/-x, where x is the number of days before/after the start day, and time is in 24-hour format. Assume standard time without daylight saving for all timezones. If seconds aren't specified, assume it to be 00.

**Model response (GPT-4o):** {"explanation": "EST is 3 hours ahead of PST. To convert 22:00 EST to PST, subtract 3 hours from 22:00. This results in 19:00 PST.", "day": "same_day", "time": "19:00:00" }

**Label:** {"day": "same_day", "time": "19:00:00"}

---

# E  QUALITY OF INSTRUCTIONS

A critical question is whether the instructions in the prompts of `ToT-Arithmetic` regarding the format of the output are of high quality and easily understood by large language models (LLMs). Notably, Gemini 1.5 Pro, GPT-4, and Claude-3-Sonnet missed the instructions only $0.05\%$, $0.65\%$, and $4.76\%$ of the time, respectively. This low error rate suggests that the instructions are of high quality. Consequently, we were able to establish a benchmark that allows us to focus purely on the temporal reasoning abilities of the models.

## F    INSTRUCTIONS TO PARTICIPANTS

For the crowd-sourcing section in creating the `ToT-Arithmetic` dataset (Expand step), we gave the following instructions to the annotators.

---

**Time Arithmetic Benchmark Compilation**

Thank you for participating in our eval hour to help us expand our dataset to cover all the categories of time arithmetic that we can think of.

**Terminology:**

- **Time arithmetic:** Calculations with time values, often involving years, months, days, hours, minutes, seconds.
- **Category:** A high-level category of time arithmetic operations, such as addition/subtraction, time conversion, etc.
- **Examples:** Real-life sentences that fall into a category. For instance, "Today is 27 July 2020 and I was told that my furniture will be delivered to me in exactly 60 days from now. On what date will the furniture be delivered?" is an example of addition.

**Goal:**    Our goal is to cover as many real-life categories and subcategories related to time arithmetic as possible. We also want each subcategory to have multiple different real-life examples.

**Levels of Importance of Contributions:**

1. Discovering/adding a new category.
2. Adding new real-life examples within a subcategory (please contribute more in less densely populated areas).

Corner cases are useful, but please don't focus all your time on them. Discovering broader categories would be the most useful!

Please try to add new examples which are as different from existing ones as possible.

Thanks!

---

## G    REPRODUCIBILITY STATEMENT

To ensure the reproducibility of our work, we provide the following resources and information:

**Benchmark creation:** A detailed description of the construction methodology for our temporal reasoning benchmark is available in Section 3. This includes the process of creation of both `ToT-Semantic` and `ToT-Arithmetic`.

**LLM access:** The LLMs evaluated in this study are publicly accessible via API calls. We specify the names of the LLMs and the corresponding versions used for our experiments in Section 4.

**Evaluation procedure:** Appendix D outlines the evaluation procedure used for our study along with some examples to better clarify the procedure.

To foster further research in this area, we are open-sourcing the datasets and evaluation framework used in our experiments: `https://huggingface.co/datasets/baharef/ToT`

## H    LIMITATION AND FUTURE WORK

The current work has several limitations that provide avenues for future research:

**Single-Sentence Time Anchoring .**    This benchmark focuses on scenarios where the start and end times of a fact are both mentioned within a single sentence. However, in real-world scenarios,

temporal information can be spread across multiple sentences or even documents. It is worth noting that this setup is easily convertible to the more general case where temporal information can be spread across multiple sentences. While we chose to focus on the single-sentence setup for this initial work, future research could readily adapt the benchmark to the multi-sentence scenario and explore the challenges and opportunities it presents.

**Exclusive Focus on Explicit Temporal Facts (By Design).** This benchmark intentionally focuses solely on explicit temporal facts (those with clear time anchors), excluding static facts (those without time anchors). This deliberate choice was made to ensure the benchmark specifically targets and assesses models' capabilities in temporal reasoning. However, future work could expand the scope to include static facts, offering a more comprehensive evaluation of both temporal and general factual reasoning.

