# OpenReview forum: "Test of Time: A Benchmark for Evaluating LLMs on Temporal Reasoning"
_ICLR.cc/2025/Conference — ICLR 2025 Poster_

### Official Review · Reviewer_SQWS · 2024-11-03

**Soundness:** 3
**Presentation:** 2
**Contribution:** 3
**Rating:** 6
**Confidence:** 3

**Summary:**

In this work, the authors introduce two novel synthetic datasets, TOT-Semantic and TOT-Arithmetic, specifically designed to evaluate LLMs’ temporal reasoning abilities with graph-like facts from two perspectives: (1) understanding the semantics and logic of time, and (2) performing accurate temporal arithmetic. The authors also conduct extensive experiments to examine how LLM performance is influenced by the graph structure, graph size, question type, and fact ordering of the problem.

**Strengths:**

The method works on temporal reasoning with LLM, an important area of research that contributes to understanding the model's overall complex reasoning capabilities.

The authors conduct several experiments. Their analysis and the data offer valuable insights for future research.

**Weaknesses:**

The paper lacks detail on dataset construction. For instance, how are the final questions generated in both TOT datasets? Are templates being used? (see also question 1)

The number of baselines is limited. Additional approaches could include directly generating code for TOT-Arithmetic or applying few-shot or self-consistency.

**Questions:**

1. Have the authors considered how the format of the date/time, such as words versus numerical format, might influence the model’s performance?

2. For 4.1, 4.1.1, what task does the author evaluate?

---

> ### Author Response · Authors · 2024-11-20
> **Thanks for your review**
>
> Thank you for the thoughtful questions. Regarding question generation, for ToT Semantic, the questions are generated using multiple templates, one for each question type. We have added Appendix B to a revision (to be uploaded within 24 hours) which includes algoboxes detailing this process. For ToT Arithmetic, the questions were mainly written by the people who helped create questions, and in some cases modified by authors to make them into full-form questions. Regarding the date/time format for ToT Arithemetic, we included many variations of both date and time in the dataset. We consider this a robustness strategy; however, we did not drill down into the empirical performance differences between date/time format.
>
> Regarding baselines, we did not consider few-shot, self-consistency, or code generation. We consider these important avenues for future modeling innovation that can make use of our ToT benchmark framework.
>
> Regarding Section 4.1 and 4.1.1, all these analyses are about ToT-Semantic.

---

> > ### Author Response · Authors · 2024-11-27
> > **Revision uploaded**
> >
> > Thanks again for your time and thoughtfulness reviewing our work. We have uploaded a revision which implements aspects of our responses. Please let us know if you have any additional questions prior to the end of the discussion period. Thank you!

---

> > > ### Comment · Reviewer_SQWS · 2024-11-30
> > >
> > > Thank you for your response. I keep my positive score and believe the score reflects the paper's quality.

---

### Official Review · Reviewer_CVKy · 2024-11-04

**Soundness:** 3
**Presentation:** 3
**Contribution:** 3
**Rating:** 8
**Confidence:** 4

**Summary:**

Dealing with the dataset quality and potential leakage problems, this paper introduces a novel method to synthesize a benchmark for comprehensive temporal reasoning benchmarks. The benchmark contains semantic and arithmetic questions with fine-grained topology control. Extensive experiments are conducted and show insightful conclusions.

**Strengths:**

- The data synthesis process benefits from the graph-guided control, and could be generalized to many other tasks.
- The constructed data are comprehensive and include many perspectives with quality control.
- Experiments are extensively conducted on multiple aspects, and provide some insights on future directions.

**Weaknesses:**

- Some claims lack of quantitative evidence:
    - “real-world data that LLMs may have encountered during pre-training or employ anonymization techniques that can inadvertently introduce factual inconsistencies” Could you add some quantitative evidence showing the GPT-4 or Gemini-1.5 Pro baselines have pre-training data contaminations?
    - “LLMs could even potentially guess the original entities due to their adjacent relations” This also lacks of quantitative evidence. If this is a commonsense, there should be relevant references cited.
- The literature review is not sufficient, and there are many researches on math-related temporal reasoning tasks. There lacks of relevant references in the introduction and the related work.
    - Wang, Y., & Zhao, Y. (2023). Tram: Benchmarking temporal reasoning for large language models. *arXiv preprint arXiv:2310.00835*.
    - Chu, Z., Chen, J., Chen, Q., Yu, W., Wang, H., Liu, M., & Qin, B. (2023). Timebench: A comprehensive evaluation of temporal reasoning abilities in large language models. *arXiv preprint arXiv:2311.17667*.
    - Su, Z., Zhang, J., Zhu, T., Qu, X., Li, J., Zhang, M., & Cheng, Y. (2024). Timo: Towards Better Temporal Reasoning for Language Models. *arXiv preprint arXiv:2406.14192*.

**Questions:**

- Some details are missing.
    - Line 212: “we generated questions per graph generation and per question type”: Please explain how to generate such questions. Are they generated from templates, manual annotations, or LLMs?
    - Line 369: Is it because the superior performance on longer contexts? Is there a correlation between long-context performance (or overall task performance e.g., MMLU, GSM8K, MATH500) and the final temporal reasoning performance? Are there sufficient test cases with more edges for providing robust evaluation?
- Typos:
    - Line 275: Funcionalizing → Functionalizing

---

> ### Author Response · Authors · 2024-11-20
> **Response**
>
> Thank you for the detailed review!  Please find our replies below:
>
> ### **Data Contamination**
>
> > Could you add some quantitative evidence showing the GPT-4 or Gemini-1.5 Pro baselines have pre-training data contaminations?
>
> This is difficult to ascertain for any proprietary LLM, however here we are referring to Temporal Reasoning tasks like TempTabQA[1] or MenatQA [2] which are based on real events (with well known entities) that are documented on resources like Wikipedia, social media, and/or the Internet at large.  These resources are well known to be used for training LLMs [3].  Using entities and relationships a LLM has seen in pre-training detracts from the ability to measure its reasoning capability.
>
> [1] https://arxiv.org/abs/2311.08002
>
> [2] https://arxiv.org/pdf/2310.05157
>
> [3] https://platform.openai.com/docs/bots
>
> > “LLMs could even potentially guess the original entities due to their adjacent relations”
>
> We are happy to remove this statement, but believe it to be true based on current capabilities of LLMs:
>
> 1. TGQA uses a table to map entities from one name to another of the same type.  This mapping is less probable than the real relationship (which the LLM has seen in its training data).
> 2. We also know that analogical reasoning is well studied in LLMs (e.g. [4]).
> 3. “De-masking” a TGQA example then, is asking a LLM to find a more likely assignment of entities to relationships than the one offered.
>
> For more information on the related task of text deanonymization, we refer to [5].
>
> [4] https://arxiv.org/pdf/2404.12728v1
> [5] https://arxiv.org/pdf/2303.12429
>
> ### **Generation Question**
> Thanks for the question, in brief:
> - The generation uses templates for each question type.
> - First a graph is sampled
> - Then facts are sampled from the graph until the proper question template can be created.
>
> We will add this to the manuscript explicitly as an algorithm block in the Appendix.
>
> ### **Related Work**
> Thank you for bringing these three papers to our attention. Each of them addresses aspects of LLM temporal reasoning distinct from the purpose of ToT:
>
>  * The questions in TRAM (Wang and Zhao, 2023) are all multiple-choice, and do not require LLMs to reason through a large list of temporal facts from a knowledge graph.
>  * The questions in TimeBench (Chu et al. 2023) are collected from ten existing real-world datasets, one of which requires reasoning through temporal facts provided in the context. ToT goes beyond such datasets by providing controllable, comprehensive temporal relationship collectinos via synthetic graph generation.
>  *  The questions in Timo (Su et al. 2024) are grouped into two categories: math-time and pure-time. The math-time category contains questions that are similar in nature to our ToT-Arithmetic dataset; however, we cover several more categories that do not seem to be included in the math-time subset of Timo. This includes questions about calendar schedules, timezones, trick questions that tend to mislead models, questions about conversions between BC and AD times, and questions that require applying multiple different operations to find the final answer. The pure-time category is in the same vein as our ToT-Semantic subset; however, they mostly focus on commonsense temporal reasoning with a focus on real-world facts and temporal knowledge, whereas our focus is mainly on multi-hop temporal reasoning on generic facts which require can be only answered through reasoning and not based on the parametric knowledge of the models. In that regard, we believe the two datasets are complementary.
>
> We will add these details to our related work section.

---

> > ### Comment · Reviewer_CVKy · 2024-11-20
> > **Thanks for your response**
> >
> > > Data contamination
> >
> > Thanks for the literature review. You don't have to remove any statements, please consider adding these explanations to the main content in the revised version.
> >
> > > Generation Question
> >
> > Thanks for the explanation. I think an algorithm block in the Appendix really helps the readers.
> >
> > > Related Work
> >
> > Thanks for the thorough comparison. But I still think the reasoning path is not affected by the multi-choice answer format. Fine-grained arithmetic subtasks are good, but there are some overlaps with other datasets. I think it would be better to include these comparisons to the main content and lower the contribution a little bit.
> >
> > ---
> >
> > Based on your comments, I'd like to raise my overall assessment. Good luck to you all.

---

> > > ### Author Response · Authors · 2024-11-27
> > > **Revision uploaded**
> > >
> > > Thanks again for your time and thoughtfulness reviewing our work. We have uploaded a revision which implements aspects of our responses. Regarding your review, we have added the additional literature reviews in the "Temporal reasoning" section (last paragraph of page 2), and we have added algoblocks detailing our question generation method for ToT-Semantic in Appendix B. Please let us know if you have any additional questions prior to the end of the discussion period. Thank you!

---

### Official Review · Reviewer_5y55 · 2024-11-04

**Soundness:** 3
**Presentation:** 3
**Contribution:** 2
**Rating:** 6
**Confidence:** 4

**Summary:**

This paper focuses on evaluating the temporal reasoning abilities of large language models (LLMs). The authors introduce a new synthetic dataset, Test of Time (ToT), which consists of two tasks: ToT-Semantic for temporal semantics and logic, and ToT-Arithmetic for temporal calculations. The study evaluates five LLMs and analyzes the impact of factors like graph structure, question type, and fact order on performance. The findings provide insights into LLMs' strengths and weaknesses in temporal reasoning.

**Strengths:**

-	The proposed ToT benchmark is designed to address the limitations of existing benchmarks by encompassing a wider variety of graph structures and question types, enabling a more nuanced evaluation of LLMs' temporal reasoning abilities
-	The authors offer an evaluation of temporal reasoning by decoupling it into semantic and arithmetic aspects. This two-pronged approach provides a more detailed analysis of LLM capabilities.

**Weaknesses:**

-	As mentioned in the limitation section, the benchmark focuses on scenarios where both the start and end times of a fact are mentioned within a single sentence. But real-world temporal information can be spread across multiple sentences or documents.
-	The authors generate questions using templates, which might not fully capture the complexity and variability of natural language found in real-world temporal reasoning tasks.

**Questions:**

1.	How would the performance of LLMs change if the benchmark included static facts in addition to explicit temporal facts?

---

> ### Author Response · Authors · 2024-11-20
> **Response**
>
> Thank you for the comments and questions!
>
> ### **Long distance temporal dependencies**
> Regarding your first comment, yes, we have explicitly disclaimed this limitation in Appendix F, noting it as an area for future work. We regret that due to space limitations, we could not explore this area further.
>
> ### **Template Generation**
> Regarding your second comment, we agree that our reliance on templates makes the examples in our benchmark potentially less-realistic than real-world examples in terms of natural language. However, this is a trade-off we make to enable us to generate highly-complex temporal and relational structures. The purpose of ToT is to isolate LLM capacity for reasoning over structured (temporal) knowledge as opposed to natural language comprehension.
>
> ### **Static and Temporal Facts**
> We do not believe that the results would change much.  Our randomly generated examples for ToT-Semantic already contain facts that vary in their duration (ie. it has both “short” and “long” facts).  A static fact would then be one with maximum length -- maintaining its truth throughout the example.  Nonetheless, we believe that the combination of static and temporal facts is an interesting area for future work.

---

> > ### Author Response · Authors · 2024-11-27
> > **Revision uploaded**
> >
> > Thanks again for your time and thoughtfulness reviewing our work. We have uploaded a revision which implements aspects of our responses. Please let us know if you have any additional questions prior to the end of the discussion period. Thank you!

---

### Official Review · Reviewer_oZRw · 2024-11-04

**Soundness:** 3
**Presentation:** 3
**Contribution:** 3
**Rating:** 8
**Confidence:** 4

**Summary:**

This paper introduces two datasets specifically crafted to evaluate large language models (LLMs) on temporal reasoning across diverse scenarios. The authors argue that existing benchmarks for temporal reasoning primarily use question-answering tasks based on Knowledge Graph -style temporal facts about well-known entities. Such benchmarks may reflect a model’s capacity to leverage prior knowledge rather than assess true temporal reasoning skills. To this end. the proposed datasets aim to measure two core temporal reasoning abilities of LLMs: (1) understanding the semantics and logic of time, and (2) performing temporal arithmetic.

**Strengths:**

- For the ToT-Semantic dataset, designed to evaluate LLMs on temporal semantics and logic, the authors employ seven graph generation algorithms and develop eight manually crafted question types. This diversity allows the generation of a large volume of synthetic questions, adding rigor to the dataset and covering various temporal reasoning facets.

- The study provides detailed insights into the temporal reasoning capabilities of frontier LLMs, including how factors such as graph size, question type, and temporal fact ordering influence performance. These observations offer valuable understanding into both the strengths and limitations of current LLMs in temporal reasoning.

**Weaknesses:**

- While ToT-Semantic focuses on temporal semantics and logical reasoning, the paper does not clearly explain how the graph generation process ensures the correctness of graph evolution. Specifically, the distinction between generating static graphs and those with temporal dynamics is not addressed, leaving questions about the dataset's fidelity to real-world temporal processes.

- In introduction, the paper emphasizes the importance of evaluating LLMs on temporal reasoning but does not clearly explain why a graph structure is essential for this assessment. Could the authors elaborate on the necessity of graphs in this context?

**Questions:**

As mentioned in weakness.

---

> ### Author Response · Authors · 2024-11-20
> **Response**
>
> Thanks for your time and thoughtful review!  Let us briefly address the questions you raised:
>
> ### **Graph Generation**
> Great question!  We agree the description of our algorithm for ensuring temporal and relational consistency was limited (explained partially at L195):
>
> > "Then, for each edge (u, v) labeled with a relation r, we assign a valid time interval [t1, t2] that respects the relation types."
>
> Our approach for ensuring overall graph correctness is similar. At each generation step -- graph generation, relation generation, time interval generation -- we check to make sure the generated nodes, edges, relations, and time intervals are consistent with the existing objects in the temporal knowledge graph. If not, we re-generate until they are, or until a pre-specified step count has been reached. If the latter, we throw out the example and start again. We will include these details in our revision.
>
> Regarding your second point about the "distinction between generating static graphs and those with temporal dynamics", we note that every temporal graph has a static *backbone* -- the structure remaining if all timestamps are removed. As discussed in S3.1, we use static graph generators to generate the backbones, and then we impute timestamps consistent with the relation types and logical rules.
>
> ### **Are Graphs Necessary?**
> To your second question, we note that every collection of temporal facts of the form (object, relation, subject, time_interval) can be expressed as a temporal graph. ToT-SEMANTIC specifically targets LLM reasoning ability over such collections. We do not claim that graph generators are the only way to construct such a benchmark. However, because all temporal fact collections contain an underlying graph, we are proposing a graph-generator-based framework to produce benchmark examples. We argue that a framework that exposes generation of the static graph backbone is more controllable and allows for a final dataset that is more comprehensive w.r.t. the variety and complexity of temporal relationships between generated entities (L49).
>
> We will extend the discussion to further motivate graph generation as a useful tool for generating different temporal dependency structures.

---

> > ### Author Response · Authors · 2024-11-27
> > **Revision uploaded**
> >
> > Thanks again for your time and thoughtfulness reviewing our work. We have uploaded a revision which implements aspects of our responses. Regarding your review, we have added additional justification for our choice of graph generation as a technique in Appendix A. Please let us know if you have any additional questions prior to the end of the discussion period. Thank you!

---

### Comment · Area_Chair_SyJD · 2024-11-26
**Response to the authors' rebuttal**

Dear reviewers, the ICLR discussion period is extended. Could you please take a look at the authors' rebuttal and other reviews, and see whether you would like to update your ratings? The authors would greatly appreciate your consideration and responses.

---

### Meta-Review · Area_Chair_SyJD · 2024-12-20

**Metareview:**

This paper introduces the Test o time benchmark to test large language models on temporal reasoning tasks. The datasets behind allow for a thorough investigation of how different factors, such as graph size, temporal factor sequence, and question type, affect LLM reasoning performance. Through this benchmark, the authors examine the strengths and weaknesses of existing LLMs in temporal reasoning. These findings can

This work receives four positive ratings (8, 8, 6, 6). The reviewers find this work is timely and important to understand LLMs' complex and temporal reasoning capabilities. The dataset and benchmark constructed can offer new understanding of both the strengths and limitations of current LLMs in temporal reasoning.

Overall, though the AC feels this work does not generate deep insights into advancing LLMs for temporal reasoning, I agree with the reviewers that this work (the benchmark/dataset constructed) is an early attempt to examine LLMs' temporal reasoning. Given four positive ratings and a successful rebuttal, an acceptance recommendation is given to this work.

**Additional Comments On Reviewer Discussion:**

This submission receives four positive ratings during the initial reviews, with three reviewers acknowledging the authors' rebuttal.

Though all positive, most these reviews/responses to rebuttal are shot and lack of detailed feedback. It would be better if the reviews/responses could help provide strong signals for the authors to further improve this work. The engagement during rebuttal was not particularly enthusiastic and informative.

---

### Decision · Program_Chairs · 2025-01-22

Accept (Poster)